# Rethinking Prompt Optimization: Reinforcement, Diversification, and Migration in Blackbox LLMs

## Abstract

An increasing number of NLP applications interact with large language models (LLMs) through black-box APIs, making prompt engineering critical for controlling model outputs. While recent Automatic Prompt Optimization (APO) methods iteratively refine prompts using model-generated feedback, *textual gradients*, they primarily focus on error correction and neglect valuable insights from correct predictions. This limits both their effectiveness and efficiency. In this paper, we propose a novel APO framework centered on enhancing the feedback mechanism. We reinterpret the textual gradient as a form of negative reinforcement and introduce the complementary positive reinforcement to explicitly preserve beneficial prompt components identified through successful predictions. To mitigate the noise inherent in LLM-generated feedback, we introduce a technique called feedback diversification, which aggregates multiple feedback signals, emphasizing consistent, actionable advice while filtering out outliers. Motivated by the rapid evolution and diversity of available LLMs, we also formalize Continual Prompt Optimization (CPO), addressing the practical challenge of efficiently migrating optimized prompts between different model versions or API providers. Our experiments reveal that naive prompt migration often degrades performance due to loss of critical instructions. In contrast, our approach consistently outperforms strong baselines, achieving significant accuracy improvements, faster convergence, and lower computational costs in both standard and migration scenarios.

## 1 Introduction

Traditionally, NLP tasks have relied on direct fine-tuning of pretrained foundation models (Bommasani et al., 2021; Devlin et al., 2019; Lewis et al., 2020; Radford et al., 2018; Raffel et al., 2020) on downstream datasets (Davari et al., 2019; Farahnak et al., 2021; Davari et al., 2020; Yang et al., 2023b; Marks et al., 2024; Davari, 2020). This process enables models to adapt their internal parameters to task-specific distributions. Parallel to fine-tuning, methods utilizing internal model representations, such as hidden states (Rogers et al., 2021; Davari et al., 2022a), gradients, and attention patterns (Kornblith et al., 2019; Raghu et al., 2021; Davari et al., 2023; 2022b), have inspired advanced optimization techniques including prompt tuning (Li & Liang, 2021; Lester et al., 2021), LoRA (Hu et al., 2022), and other parameter-efficient methods (Davari & Belilovsky, 2024b; Yadav et al., 2023; Davari & Belilovsky, 2024a; Yu et al., 2024).

However, the NLP landscape is shifting toward closed-weight Large Language Models (LLMs) accessed via black-box APIs, where internal representations and gradients are inaccessible. Consequently, traditional fine-tuning and interpretability-driven methods become impractical, placing greater emphasis on prompt design as the primary mechanism for model adaptation. The increasing reliance on API-based models, such as GPT-3 and GPT-4, has spurred a wave of commercial applications (OpenAI, 2023; Bubeck et al., 2023), where carefully engineered prompts critically influence model outputs (Luo et al., 2022; Kojima et al., 2022; Wang et al., 2022; Zhou et al., 2022a; Madaan et al., 2023; Bai et al., 2022; Chen et al., 2023c). Yet manually formulating effective prompts is costly, requiring extensive domain knowledge, prompting expertise (Jiang et al., 2023b; Wei et al., 2022; Kong et al., 2023), and considerable trial-and-error effort.

To alleviate these challenges, recent work has introduced Automatic Prompt Optimization (APO) methods (Wang et al., 2023a; Yang et al., 2023a; Zhou et al., 2022b; Pryzant et al., 2023), which iteratively refine prompts based on model performance feedback, typically measured on a held-out validation set. These refinement methods either follow predefined actions such as deleting, paraphrasing, or rearranging prompt elements (Zhou et al., 2022b; Schnabel

& Neville, 2024), or employ a secondary LLM to generate candidate prompts guided by model predictions and corresponding feedback signals (Wang et al., 2023a; Pryzant et al., 2023; Yang et al., 2023a). Such feedback, often referred to as the *textual gradient* (Wang et al., 2023a), is typically generated from incorrect predictions and serves as a form of negative reinforcement aimed at correcting errors. While useful, this singular focus neglects valuable information available from correct predictions, limiting both efficiency and effectiveness.

In this paper, we enhance the feedback component of prompt optimization, a dimension relatively underexplored compared to search and planning approaches explored in prior work (Wang et al., 2023a; Pryzant et al., 2023; Schnabel & Neville, 2024). Our insights complement existing methods and can be seamlessly integrated with them. Specifically, we reconceptualize the textual gradient as *negative reinforcement* (Mnih et al., 2013), capturing corrective signals from incorrect predictions. To augment this, we introduce *positive reinforcement*, explicitly identifying and reinforcing beneficial prompt components discovered through successful predictions.

Additionally, to manage variability and noise in LLM-generated feedback, we propose *feedback diversification*. This method aggregates multiple feedback samples (both positive and negative) for the same prompt and training examples, summarizing them via an LLM. The aggregation leads to an emphasis on consistent, impactful instructions while filtering out noisy outliers. This process resembles the peer review process, where diverse perspectives collectively highlight the most valuable insights. We combine these concepts into **BReAD**, short for **B**alanced **Re**inforcement and **A**ggregated **D**iversification, a unified framework yielding robust, efficient prompt optimization.

Prompt optimization becomes even more challenging when migrating optimized prompts between different LLMs or across API providers. Rapid advancements in LLM technology frequently necessitate adapting existing optimized prompts to newer models, such as from GPT-3 to GPT-4. To formally address this scenario, we introduce *Continual Prompt Optimization (CPO)*, inspired by the machine learning paradigm of continual learning (Parisi et al., 2019; Li & Hoiem, 2017; Davari et al., 2022a). Similar to continual learning, CPO emphasizes retaining critical knowledge, specifically valuable prompt elements, while adapting to new models.

Naive prompt migration strategies, involving direct reuse of previously optimized prompts, frequently degrade performance due to differences in model capabilities, tasks, and providers (Kojima et al., 2022; Zhou et al., 2022b; Zhang et al., 2022; Ma et al., 2023; Chen et al., 2023a). Standard APO methods can exacerbate this issue, inadvertently discarding or altering crucial prompt elements during iterative optimization. Our proposed framework mitigates this problem by explicitly preserving essential instructions through balanced reinforcement and feedback diversification, ensuring efficient and effective prompt migration.

Empirical evaluations demonstrate our framework consistently surpasses strong baselines in standard optimization and realistic migration scenarios. Experiments migrating prompts from `GPT-3.5-turbo` to `GPT-4o` show significantly improved accuracy and reduced computational cost through accelerated convergence. Overall, our contributions are as follows:

1. We introduce and formalize *Continual Prompt Optimization (CPO)*, a framework designed to efficiently adapt optimized prompts across evolving LLM versions, effectively addressing performance degradation and instruction loss common in naive migration approaches.

2. We propose **BReAD**, an APO method combining structured positive/negative reinforcement and feedback diversification, achieving robust accuracy improvements, faster convergence, and reduced API usage.

## 2 Related Work

Automatic prompt optimization (APO) methods can be broadly categorized based on their level of access to model internals. The first category includes approaches that assume full or partial access to internal model states, such as parameters, gradients, or output probabilities. These methods are primarily applicable to open-source models like LLaMA (Touvron et al., 2023a;b; Grattafiori et al., 2024) or Mistral (Jiang et al., 2023a; team, 2024). Leveraging this internal information, these methods can directly train additional parameters, such as soft prompts (Li & Liang, 2021; Lester et al., 2021; Hu et al., 2022; Wang et al., 2023b; Qin & Eisner, 2021), or optimize discrete prompts using gradient-guided search (Shin et al., 2020; Wen et al., 2023; Gao et al., 2020; Chen et al., 2023b). Some approaches even involve training the prompt generator itself (Hao et al., 2023; Wang et al., 2022). However, these techniques are not feasible when interacting with black-box APIs, which is the primary focus of our paper.

The second category encompasses methods designed for black-box interaction, where models are accessed only through APIs without internal visibility. Black-box APO methods generally fall into two subcategories: (1) iterative generation methods and (2) search and planning-based methods.

**Iterative Generation and Evaluation**   Iterative generation methods repeatedly propose new prompt candidates, evaluate their effectiveness based on the model's performance, and select the best-performing prompts for subsequent iterations. These methods typically rely on performance-driven cycles of generation and evaluation, often using external metrics or validation sets.

Automatic Prompt Engineer (APE)(Zhou et al., 2022b), for example, iteratively generates prompt variations through semantic modifications of an initial prompt, selecting the best candidate based on validation accuracy until convergence. Similarly, Optimization by PROmpting (OPRO)(Yang et al., 2023a) employs an LLM to iteratively propose new prompts informed by previous prompts and their respective performance metrics. Unlike APE, OPRO can generate substantially different prompts rather than mere semantic variations.

Despite their effectiveness, iterative methods like APE and OPRO primarily rely on implicit and unstructured feedback signals derived indirectly from performance metrics. Consequently, these methods are limited in the depth of optimization they can achieve. In contrast, our proposed approach explicitly incorporates structured feedback, both positive and negative, derived directly from model predictions, significantly enriching the feedback mechanism and enhancing the optimization process.

**Search and Planning-based Approaches**   Another family of black-box APO methods frames prompt optimization as a search or planning problem. These methods leverage systematic exploration techniques, such as tree search or multi-objective optimization, to efficiently navigate the prompt space. Prominent examples include PromptAgent (Wang et al., 2023a), ProTeGi (Pryzant et al., 2023), and SAMMO (Schnabel & Neville, 2024).

PromptAgent employs Monte Carlo Tree Search (MCTS) (Coulom, 2006) to formulate prompt optimization as sequential decision-making, guided by a textual gradient derived from incorrect predictions. Similarly, ProTeGi uses beam search to systematically explore prompt modifications, also relying primarily on corrective textual gradients. In contrast, SAMMO applies a multi-objective optimization framework, performing structural modifications, such as adding, removing, or replacing prompt components, guided by predefined objectives.

While these search-based methods effectively explore the prompt space, they predominantly utilize corrective feedback from incorrect model predictions, neglecting beneficial insights from correct predictions. Our proposed framework complements these methods by enhancing the feedback component. By explicitly integrating structured positive and negative reinforcement signals, along with feedback diversification to manage feedback noise, our approach improves both efficiency and robustness. Importantly, our framework can be seamlessly combined with these existing search-based methods, potentially increasing their performance by enriching their feedback mechanisms.

## 3   Methodology

In this section, we introduce our proposed framework for APO, which is composed of five primary modules (see Figure 1).

**Forward Generation:**   This module generates model predictions by applying the current prompt to a batch of training examples via the underlying LLM.

**Evaluation:**   The evaluation module measures the effectiveness of a given prompt based on the accuracy of the model's predictions on a held-out validation set. The evaluation can be deterministic, such as measuring accuracy via the exact match between the model's predictions and the ground truth labels, or it can be LLM-based, where the model is asked to evaluate the quality of the predictions. In this paper, we focus on deterministic evaluation, but the framework can be easily extended to include LLM-based evaluations.

**Feedback Generation:**   This module is responsible for producing structured feedback signals based on the model's predictions using the current prompt. It generates positive feedback, which explicitly identifies and encourages retaining prompt components that lead to correct model predictions, and negative feedback, which highlights potential issues that lead to incorrect predictions, suggesting specific modifications to improve model accuracy. When employing feedback diversification, this module generates multiple feedback signals (both positive and negative) for the same prompt and

training batch. These signals are then aggregated via a summarization call to an LLM, resulting in a consolidated feedback signal emphasizing consistent and impactful instructions while minimizing noise.

**Prompt Update:** This module generates a new prompt based on the current prompt, the training batch processed by the model, and the feedback signals generated in the previous step.

**Search Module (Abstracted):** The search module proposes the next candidate prompt to be further explored. The suggestion is based on the evaluation metrics and other search-related criteria (e.g., visitation frequency in MCTS). Since this research focuses explicitly on feedback enhancement rather than search or planning techniques, this module is treated as an abstract, interchangeable component that can adopt existing APO methods, including PromptAgent (Wang et al., 2023a), ProTeGi (Pryzant et al., 2023), or iterative generation methods such as APE (Zhou et al., 2022b) and OPRO (Yang et al., 2023a).

In our experimental evaluation, Section 4, we specifically utilize PromptAgent (Wang et al., 2023a) as the baseline search and planning method due to its established effectiveness and robust performance across diverse prompt optimization tasks (Zhang et al., 2025; Li et al., 2025). Nevertheless, the modular design of our feedback generation component allows straightforward integration with other existing APO frameworks, making our approach versatile and broadly applicable.

## 4 Experiments

### 4.1 Data, Metrics, and Models

We evaluate our method across five tasks that collectively span causal, spatial, tabular, inferential, and semantic reasoning, covering both classification and regression settings. Three of these tasks are drawn from the BBH benchmark (Suzgun et al., 2022), a widely used suite in prompt optimization (Schnabel & Neville, 2024; Wang et al., 2023a; Zhou et al., 2022b; Pryzant et al., 2023; Yang et al., 2023a) and prompt engineering research (Wei et al., 2022; Kojima et al., 2022; Wang et al., 2022; Zhou et al., 2022a; Madaan et al., 2023; Bai et al., 2022; Chen et al., 2023c). We focus on a representative subset of BBH tasks selected for their diversity in reasoning requirements: Causal Judgment evaluates causal inference through binary classification; Geometric Shapes assesses geometric and spatial reasoning by requiring the model to interpret SVG path strings in a multiclass classification setting; and Penguins tests the model's ability to perform table-based classification by reasoning over structured data.

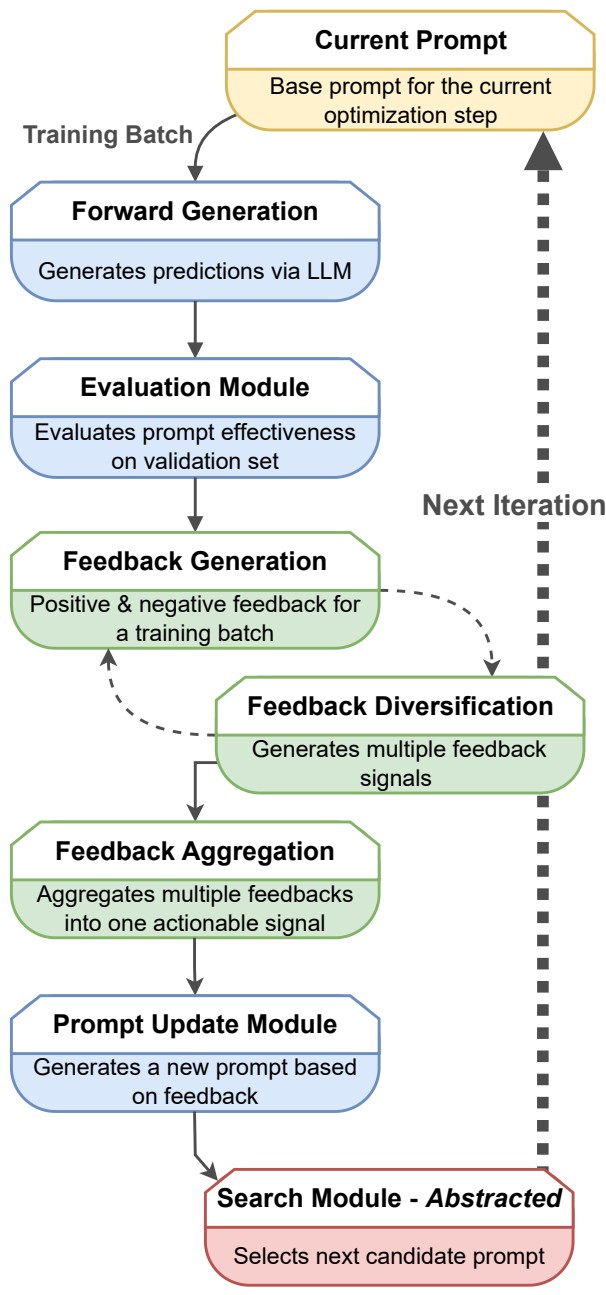

Figure 1: Overview of our proposed framework for automatic prompt optimization (APO). The framework consists of five primary modules: Forward Generation, Evaluation, Feedback Generation, Prompt Update, and Search Module. The Search Module is abstracted to allow for the integration of various search and planning methods.

To complement these, we include two additional benchmarks outside of BBH. The CommitmentBank (CB) (De Marneffe et al., 2019) is a natural language inference dataset in which models classify contextualized sentence pairs as entailment, contradiction, or neutral. Biosses (Soğancıoğlu et al., 2017) is a biomedical dataset that measures sentence-level semantic similarity, framed as a regression task with continuous similarity scores. This combination of tasks enables a

comprehensive evaluation of our framework's generalizability, robustness, and performance across a broad spectrum of reasoning types, domains, and output formats. For details on the dataset statistics, see Appendix B.

We use accuracy as the primary evaluation metric. During the optimization process, model performance is assessed on a held-out validation set, and final results are reported on the test set. Both GPT-3.5-turbo and GPT-4o are evaluated using the dataset's default prompts, as well as the optimized prompts generated by our proposed approach and the baseline method, PromptAgent (Wang et al., 2023a).

## 4.2 Standard Prompt Optimization (GPT-3.5-turbo)

In this section, we evaluate our methodology in the standard Automatic Prompt Optimization (APO) setting, where the goal is to improve a prompt for a fixed language model, GPT-3.5-turbo. We initialize the optimization with a task-specific *default prompt*, which is a concise, single-sentence instruction describing the task (e.g., "Answer questions about causal attribution" for Causal Judgment). The full list of default prompts is provided in Appendix C.

Each experiment is run for up to 15 optimization iterations. During this process, the feedback diversification module samples six feedback signals per batch. Positive feedback is introduced after the third or fourth iteration, depending on the dataset, as determined empirically (see Section 4.4 for details). This staged introduction ensures that early updates prioritize error correction before preserving beneficial prompt components.

Table 1 summarizes our experimental results, averaged across five random seeds. Compared to the PromptAgent baseline (Wang et al., 2023a), our approach achieves consistent and statistically significant improvements in accuracy across all tasks, ranging from 4.9% to 21.5% ($p < 0.01$). Additionally, incorporating feedback diversification, either to the baseline or in our approach, reduces the standard deviation, which highlights the role of feedback diversification in stabilizing the optimization process. Beyond accuracy, our method offers improved efficiency. By reinforcing helpful prompt elements early, it avoids redundant removals and rediscoveries, leading to faster convergence. This translates into a reduction ranging from 0.5% to 3.3% in the number of LLM calls across tasks compared to the baseline (details in Appendix D). Such efficiency is particularly valuable in real-world applications, where API costs and latency are critical considerations.

## 4.3 Prompt Migration: GPT-3.5-turbo →GPT-4o

Having established the effectiveness of our method in the standard optimization setting, we now turn to the more challenging and practically important scenario of prompt migration. This setting involves transferring prompts optimized for one language model (e.g., GPT-3.5-turbo) to a newer or different model (e.g., GPT-4o), with the goal of preserving both effectiveness and key instructional components.

We begin by evaluating the direct transferability of prompts optimized for GPT-3.5-turbo. We refer to these as expert prompts, which are the final outputs of the APO procedure described in Section 4.2. These prompts are applied directly to GPT-4o and compared against the task-specific default prompts. A full list of default and expert prompts is provided in Appendix C. As shown in Table 2, expert prompts generally outperform default prompts when transferred directly, yielding initial accuracy gains of 1.3% to 3.3% across tasks.

However, applying standard APO to these transferred expert prompts often results in worse or statistically insignificant performance improvements, compared to optimizing from the default prompt. This degradation occurs because early-stage corrective feedback can overwrite useful instructions of the expert prompt that were critical for task success. While some of this content may be recovered in later iterations, the optimization becomes inefficient and unstable. These findings highlight a key limitation of naive prompt migration: it risks discarding task-relevant knowledge embedded in prompts tuned for the source model.

Next, we evaluate the effectiveness of our framework in the prompt migration setting. The optimization process is initialized with expert prompts derived from GPT-3.5-turbo optimization (see Section 4.2) and applied to GPT-4o for up to 15 iterations. To account for the higher API cost and greater accuracy of GPT-4o, we reduce the number of feedback diversification samples to two per iteration. Unlike in the standard APO setting, where the initial prompt contains minimal task-specific guidance, we introduce positive feedback from the first iteration to preserve valuable information already present in the expert prompts. These design choices are guided by preliminary experiments, which showed diminishing returns from additional diversification and performance degradation when delaying positive feedback.

Table 1: Performance of prompt optimization methods on `GPT-3.5-turbo`, averaged over five random seeds. Results show mean accuracy (± standard deviation), with statistical significance assessed via two-tailed paired $t$-tests ($p$-value, Cohen's $d$). +FD = Feedback Diversification; +PR = Positive Reinforcement; BReAD = full method combining both. Init. Acc. refers to the accuracy achieved by the default prompt before optimization. All methods use 15 iterations and 6 feedback samples per step; positive reinforcement is introduced at iteration 3 for Casual Judgment, Geometric Shapes, and Penguins, and at iteration 4 for Biosses and CB. BReAD consistently outperforms the baseline and its variants, achieving significant accuracy gains (4.9%–21.5%) and improved stability.

| Dataset (Init. Acc.) | Method | Accuracy | $p$-value | Cohen's $d$ |
|---|---|---|---|---|
| Causal Judgment ($56.5_{\pm 3.67}$) | Baseline | $58.6_{\pm 3.98}$ | – | – |
| | Baseline + FD[*] | $60.8_{\pm 2.38}$ | 0.020 | 1.687 |
| | Baseline + PR[*] | $63.6_{\pm 2.62}$ | 0.040 | 1.336 |
| | BReAD[**] | $\mathbf{64.4}_{\pm 2.16}$ | 0.008 | 2.162 |
| Geometric Shapes ($32.7_{\pm 2.04}$) | Baseline | $52.1_{\pm 4.94}$ | – | – |
| | Baseline + FD[*] | $57.8_{\pm 3.15}$ | 0.032 | 1.439 |
| | Baseline + PR[*] | $61.6_{\pm 4.18}$ | 0.035 | 1.399 |
| | BReAD[**] | $\mathbf{63.3}_{\pm 1.16}$ | 0.004 | 2.644 |
| Penguins ($60.5_{\pm 4.87}$) | Baseline | $65.1_{\pm 4.96}$ | – | – |
| | Baseline + FD[*] | $66.1_{\pm 2.42}$ | 0.083 | 1.148 |
| | Baseline + PR[*] | $66.9_{\pm 3.97}$ | 0.043 | 1.464 |
| | BReAD[**] | $\mathbf{68.6}_{\pm 1.87}$ | 0.007 | 2.556 |
| Biosses ($25.2_{\pm 3.84}$) | Baseline | $62.5_{\pm 4.19}$ | – | – |
| | Baseline + FD[*] | $67.0_{\pm 2.92}$ | 0.044 | 1.456 |
| | Baseline + PR[*] | $68.2_{\pm 3.02}$ | 0.021 | 1.844 |
| | BReAD[**] | $\mathbf{70.4}_{\pm 2.02}$ | 0.006 | 2.654 |
| CB ($68.5_{\pm 4.22}$) | Baseline | $81.7_{\pm 3.17}$ | – | – |
| | Baseline + FD[*] | $84.2_{\pm 2.02}$ | 0.032 | 1.610 |
| | Baseline + PR[*] | $84.2_{\pm 3.73}$ | 0.049 | 1.402 |
| | BReAD[**] | $\mathbf{85.7}_{\pm 3.54}$ | 0.008 | 2.495 |

As shown in Table 3, our method consistently improves performance across all tasks, with relative accuracy gains ranging from 3.5% to 16.0% over the PromptAgent baseline ($p < 0.01$). The most substantial improvements are observed in Geometric Shapes (12.5%) and Biosses (16.0%), where preserving domain-specific instructions is especially important. In Geometric Shapes, spatial reasoning patterns embedded in the prompt are critical; in Biosses, biomedical terminology and precise phrasing directly affect model output. These results demonstrate the framework's ability to maintain and adapt task-relevant instructions during migration.

In addition to accuracy improvements, our method enhances both stability and efficiency. As in the standard APO setting, we again observe that feedback diversification reduces variance across runs, leading to more consistent convergence. Moreover, the total number of LLM calls is reduced by 4.2% to 6.2% compared to the baseline, offering meaningful cost savings given the higher API expense of `GPT-4o`. A detailed breakdown of LLM usage is provided in Appendix D.

## 4.4 Ablation Studies

In this section, we analyze the contribution of key components and hyperparameters in our framework through a series of ablation studies. All experiments are conducted using the `GPT-3.5-turbo` model with a maximum of 8 optimization iterations, starting from the minimal instruction set defined in the default prompts (see Appendix C). We report relative accuracy improvements over the PromptAgent baseline (Wang et al., 2023a), shown as a dotted red line in Figures 2a and 2b.

**Impact of Positive and Negative Reinforcement** Our framework uses structured reinforcement signals derived from model predictions. *Negative reinforcement*, based on incorrect predictions, guides the prompt toward correcting errors. *Positive reinforcement*, drawn from correct predictions, encourages the retention of effective prompt components.

Table 2: Transferability of Expert Prompts (EP) (optimized prompts from `GPT-3.5-turbo` to `GPT-4o`. Accuracy (mean ± std, over five seeds) is reported for the Default Prompt (DP) and the transferred EP at two stages: *Initial*: direct transfer without further optimization, and *Final*: after 15 iterations of PromptAgent on `GPT-4o`. Significance is assessed with two-tailed paired $t$-tests. Expert prompts yield small but significant gains on direct transfer (1.3%–3.3% improvement), yet re-optimizing them often erodes or reverses this advantage.

| Dataset | Stage | DP Acc. | EP Acc. | $p$-value | Cohen's $d$ |
|---|---|---|---|---|---|
| Causal Judgment | Initial | 71.8 ± 1.92 | **74.2** ± 3.46 | 0.033* | 1.434 |
|  | Final | 73.4 ± 1.82 | **73.8** ± 1.79 | 0.803 | 0.119 |
| Geometric Shapes | Initial | 54.8 ± 1.89 | **58.2** ± 2.22 | 0.001** | 4.138 |
|  | Final | **79.0** ± 5.32 | 75.1 ± 5.80 | 0.040* | −1.344 |
| Penguins | Initial | 92.9 ± 1.85 | **95.8** ± 1.72 | 0.025* | 1.745 |
|  | Final | **95.2** ± 2.03 | 92.3 ± 2.89 | 0.005** | −2.771 |
| Biosses | Initial | 69.9 ± 2.73 | **72.2** ± 1.78 | 0.0125* | 2.159 |
|  | Final | **76.7** ± 2.84 | 76.1 ± 1.97 | 0.381 | −0.600 |
| CB | Initial | 79.3 ± 1.57 | **80.3** ± 1.54 | 0.005** | 2.855 |
|  | Final | **80.0** ± 4.62 | 78.7 ± 2.13 | 0.381 | −0.492 |

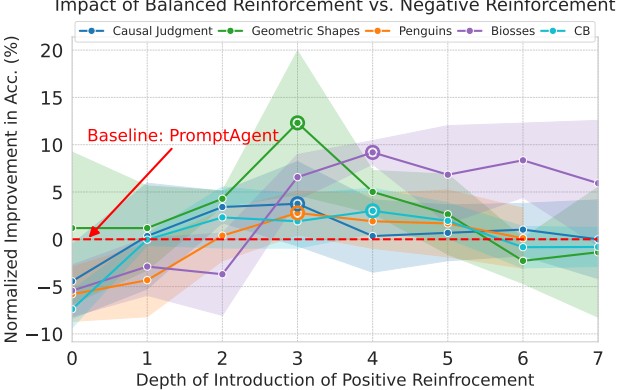

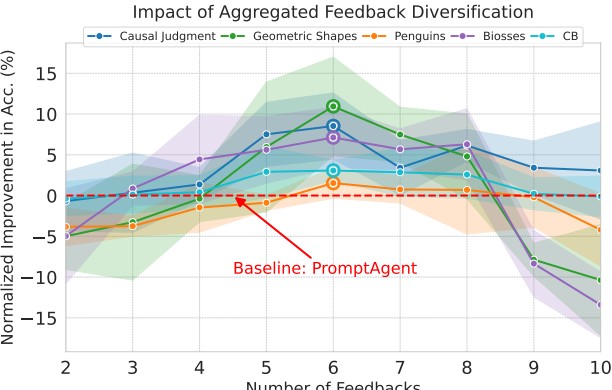

(a) Accuracy gains relative to the baseline (no positive reinforcement) when positive reinforcement is introduced at different optimization depths. Performance peaks at depth 3–4. Early introduction (depth 0–1) impedes prompt refinement, while later introduction (depth ≥ 5) leads to diminishing returns due to premature loss of useful instructions.

(b) Accuracy improvements relative to the baseline (without feedback diversification) as a function of the number of diversification samples. Performance consistently improves up to six samples across tasks, after which accuracy declines. This drop is attributed to overly generalized feedback aggregation, which dilutes specific and actionable suggestions.

A key consideration here is when to introduce positive reinforcement. If applied too early, when the prompt is still underdeveloped, it may preserve premature structures and impede beneficial edits. If applied too late, valuable components may already have been discarded, requiring inefficient rediscovery. The timing, or *depth*, of introducing positive reinforcement is therefore critical. Figure 2a shows the relative accuracy gains across tasks as a function of this introduction depth. Early introduction (depth 0 or 1) leads to underperformance across all tasks. Peak performance is achieved at depth 3 for Causal Judgment, Geometric Shapes, and Penguins, and at depth 4 for CB and Biosses. Beyond these points, performance declines as the optimization process continues to discard and re-learn useful instructions for too long before reinforcement stabilizes them. These results suggest that a brief initial phase of corrective-only feedback allows the model to build a minimal set of effective instructions, which can then be reinforced and refined in subsequent iterations.

**Impact of Feedback Diversification** We also study the role of *feedback diversification*, which aggregates multiple independently sampled feedback signals into a consolidated actionable summary. Inspired by the peer-review process, this mechanism aims to highlight consistent, high-value suggestions while down-weighting noisy or inconsistent

Table 3: Prompt migration results from `GPT-3.5-turbo` to `GPT-4o`. Each block reports the initial accuracy (± std.) of the Default Prompt (DP) and Expert Prompt (EP), followed by final post-optimization accuracy under different methods, averaged over five random seeds. All experiments start from the EPs previously optimized for `GPT-3.5-turbo`. Baseline refers to the original PromptAgent; +FD augments the baseline with Feedback Diversification; +PR adds Positive Reinforcement; BReAD combines both. Two-tailed paired $t$-test $p$-values and Cohen's $d$ measure statistical significance relative to the baseline. Significant improvements: $^*p < 0.05$, $^{**}p < 0.01$. Our method, BReAD, consistently achieves the highest final accuracy and strongest effect sizes, with gains up to 16.0% over the baseline, underscoring its robustness in real-world migration scenarios where preserving critical prompt elements is essential.

| Dataset (Init. Acc.) | Method | Final Acc. | $p$-value | Cohen's $d$ |
|---|---|---|---|---|
| Causal Judgment
DP: 71.8 ± 1.92
EP: 74.2 ± 3.46 | Baseline | 73.8 ± 1.79 | – | – |
| | Baseline + FD$^*$ | 74.7 ± 1.44 | 0.053 | 1.214 |
| | Baseline + PR$^*$ | 75.8 ± 1.78 | 0.047 | 1.265 |
| | BReAD$^{**}$ | **76.4** ± 1.59 | 0.007 | 2.280 |
| Geometric Shapes
DP: 54.8 ± 1.89
EP: 58.2 ± 2.22 | Baseline | 75.1 ± 5.80 | – | – |
| | Baseline + FD$^*$ | 79.4 ± 2.92 | 0.016 | 1.782 |
| | Baseline + PR$^*$ | 81.7 ± 3.07 | 0.021 | 1.641 |
| | BReAD$^{**}$ | **84.5** ± 2.33 | 0.008 | 2.175 |
| Penguins
DP: 92.9 ± 1.85
EP: 95.8 ± 1.72 | Baseline | 92.3 ± 2.89 | – | – |
| | Baseline + FD$^*$ | 94.2 ± 1.33 | 0.083 | 1.148 |
| | Baseline + PR$^*$ | 96.7 ± 0.88 | 0.041 | 1.493 |
| | BReAD$^{**}$ | **98.0** ± 0.73 | 0.008 | 2.449 |
| Biosses
DP: 69.9 ± 2.73
EP: 72.2 ± 1.78 | Baseline | 76.1 ± 1.97 | – | – |
| | Baseline + FD$^*$ | 78.4 ± 1.66 | 0.043 | 1.461 |
| | Baseline + PR$^*$ | 83.7 ± 3.86 | 0.003 | 3.186 |
| | BReAD$^{**}$ | **88.3** ± 2.00 | 0.0001 | 8.696 |
| CB
DP: 79.3 ± 1.57
EP: 80.3 ± 1.54 | Baseline | 78.7 ± 2.13 | – | – |
| | Baseline + FD$^*$ | 82.7 ± 2.31 | 0.029 | 1.659 |
| | Baseline + PR$^*$ | 85.3 ± 4.49 | 0.014 | 2.047 |
| | BReAD$^{**}$ | **87.5** ± 3.56 | 0.006 | 2.713 |

signals. To isolate this effect, we apply feedback diversification to the PromptAgent baseline using only negative reinforcement. Figure 2b reports relative accuracy improvements as the number of feedback samples per batch increases from 2 to 10. With just two samples, the benefits are limited, either due to insufficient variety or aggregation of conflicting signals into generic summaries. Accuracy improves steadily with more samples and plateaus around six for all datasets. Beyond this, performance drops. Manual inspection reveals that excessive diversification (8–10 samples) produces overly generalized summaries that dilute actionable guidance. These findings highlight the need for a balanced sampling strategy that captures meaningful feedback diversity without introducing aggregation noise. While our implementation uses LLM-based summarization, alternative aggregation methods such as concatenation or voting may offer complementary trade-offs. However, these approaches may increase context length and computational cost, and we leave their exploration to future work.

## 5 Conclusion and Future Work

In this work, we introduced a novel framework for Automatic Prompt Optimization (APO) specifically designed to enhance the feedback generation process when optimizing prompts for black-box Large Language Models (LLMs). Unlike prior APO methods, which predominantly rely on corrective feedback from incorrect predictions, our method integrates structured reinforcement signals derived from both correct (positive reinforcement) and incorrect (negative reinforcement) model predictions. Additionally, we introduced *feedback diversification*, an effective technique that aggregates multiple feedback signals to highlight consistently impactful instructions and reduce the influence of noisy or irrelevant feedback.

Motivated by real-world industry scenarios, we also formalized the concept of *Continual Prompt Optimization (CPO)*, addressing the practical need for efficiently adapting optimized prompts across evolving LLM versions or migrating prompts between different API providers. Our empirical evaluations demonstrated that naive migration strategies degrade performance due to inadvertent loss or distortion of crucial instructions. In contrast, our reinforcement-driven APO framework consistently outperformed the state-of-the-art baseline, increased robustness, converged faster, and reduced computational cost.

These contributions open several promising directions for future work. Exploring alternative aggregation techniques, such as weighted voting or prompt-based structuring of feedback inputs, may enhance the granularity and reliability of diversified signals. For instance, using consistent prompting templates or role-based formulations could encourage clearer, more targeted feedback from the model. Extending the framework to dynamic or interactive settings, including multi-turn dialogue, real-time systems, or multimodal inputs, could further broaden its applicability. Finally, incorporating adaptive reinforcement schedules that respond to uncertainty or performance drift may lead to more sample-efficient and resilient optimization strategies in real-world deployments.

## 6 Limitations

Despite the consistent improvements observed in prompt optimization efficiency, robustness, and migration performance, our framework has several limitations. First, our experiments are limited to GPT-family models (`GPT-3.5-turbo` and `GPT-4o`), which may constrain generalizability to other LLMs or commercial APIs. Future work should evaluate the framework's adaptability across a broader range of models and providers.

Second, the effectiveness of both reinforcement and feedback diversification strategies depends on careful hyperparameter selection, such as the number of diversification samples and the timing of positive reinforcement. While we conducted ablation studies to characterize these choices, more systematic or automated tuning may be necessary for optimal performance across diverse tasks.

Finally, although our method reduces computational cost relative to a strong baseline, it still involves multiple API calls per optimization step. This can be a limiting factor in large-scale or latency-sensitive deployments. Exploring approaches such as adaptive sampling, caching strategies, or lightweight aggregation mechanisms could help reduce this cost and further improve real-world feasibility.

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

## A Appendix

## B Dataset

Table 4 summarizes the dataset splits used in our experiments, including the number of training, validation, and test instances per task. Each dataset is designed to probe distinct reasoning capabilities of large language models (LLMs), and collectively they offer a diverse evaluation ground for prompt optimization.

**Causal Judgment** This dataset tests causal attribution skills by presenting real-world scenarios and asking whether one event caused another. It evaluates the model's ability to perform commonsense reasoning under ambiguity. Below is an example instance from the dataset:

```
Joe was feeling quite dehydrated, so he stopped by the local smoothie shop to buy the largest sized
drink available. Before ordering, the cashier told him that the Mega-Sized Smoothies were now one
dollar more than they used to be. Joe replied, "I don't care if I have to pay one dollar more, I just
want the biggest smoothie you have." Sure enough, Joe received the Mega-Sized Smoothie and paid one
dollar more for it. Did Joe intentionally pay one dollar more?
Label: Yes
```

**Geometric Shapes** A synthetic visual-reasoning-inspired dataset that describes a geometric shape via its SVG representation. The task tests the model's ability to infer spatial and comparative relationships. An example is shown below:

```
This SVG path element <path d=M 59.43,52.76 L 75.49,27.45 L 54.92,4.40 M 54.92,4.40 L 23.70,7.77 L
15.15,42.15 L 34.51,57.44 L 59.43,52.76/> draws a
Label: hexagon
```

**Penguins** This dataset examinsines the model's ability to reason about tabular data. At each instance, the model is presented with a question about penguins in a table format, and it must select the correct answer from a set of choices.

```
Here is a table where the first line is a header and each subsequent line is a penguin:

name, age, height (cm), weight (kg)
Louis, 7, 50, 11
Bernard, 5, 80, 13
Vincent, 9, 60, 11
Gwen, 8, 70, 15
For example: the age of Louis is 7, the weight of Gwen is 15 kg, the height of Bernard is 80 cm.
Question: What is the height of Gwen?
Options: A) 50 cm, B) 80 cm, C) 70 cm, D) 60 cm

Label: C) 70 cm
```

**Biosses** This biomedical sentence similarity dataset presents pairs of scientific statements and asks for semantic similarity on a scale, assessing the model's ability to reason about specialized, domain-specific language. A sample pair is shown below:

```
S1: It has recently been shown that Craf is essential for Kras G12D-induced NSCLC.
S2: It has recently become evident that Craf is essential for the onset of Kras-driven non-small cell
lung cancer.

Label: Similar
```

**CB (CommitmentBank)** CB is a SuperGLUE (Wang et al., 2019) NLI dataset designed to evaluate pragmatic inference and speaker commitment in naturally occurring sentences. It differs from standard NLI datasets because

Table 4: Dataset splits used for prompt optimization and evaluation. The training set is used during prompt updates, the validation set is used to select the best-performing prompt, and the test set is reserved for final accuracy reporting.

| Dataset | Train | Validation | Test |
|---|---|---|---|
| Causal Judgment | 30 | 60 | 100 |
| Geometric Shapes | 50 | 100 | 200 |
| Penguins | 30 | 40 | 79 |
| Biosses | 30 | 30 | 40 |
| CB | 30 | 95 | 56 |

each hypothesis is derived directly from the premise's embedded clause, minimizing annotation artifacts. Below is a representative example:

```
Premise: Some of them, like for instance the farm in Connecticut, are quite small. If I like a place
I buy it. I guess you could say it's a hobby."
Hypothesis: Buying places is a hobby.

Label: Entailment
```

In this case, the hypothesis is the complement of the clause-embedding verb say, and models must correctly infer that the sentence author is committed to the embedded proposition. This task hinges on understanding modality, clause embedding, and speaker stance, rather than surface-level lexical overlap.

These datasets collectively test a broad spectrum of reasoning abilities, ranging from causal inference and visual abstraction to factual recall, biomedical semantics, and logical entailment, making them suitable benchmarks for evaluating the generality and robustness of prompt optimization methods.

## C   Task Prompts

Each task begins with a minimalist *base prompt* that serves as the initialization point for prompt optimization. These prompts are written as system messages in the GPT-3.5-turbo and GPT-4o chat interfaces, and are intentionally kept simple to avoid embedding task-specific strategies or heuristics. The goal is to provide just enough instruction for the model to attempt the task, allowing the optimization process to refine and expand the prompt effectively. Below, we list the base prompts used for each task.

**Task: Causal Judgment**
Answer questions about causal attribution

**Task: Geometric Shapes**
Name geometric shapes from their SVG paths

**Task: Penguins**
Answer questions about a table of penguins and their attributes

**Task: Biosses**
Decide if these two sentences are (A) Not similar (B) Somewhat similar (C) Similar.

> **Task: CB**
> What is the relationship between the following premise and the hypothesis?
> Options:
> - Contradiction
> - Neutral
> - Entailment

In Section 4.2, we described how each base prompt is optimized using our reinforcement-based approach. Below, we include the resulting **expert prompts** obtained from the final iteration of the standard prompt optimization process. These reflect the outcome of the optimization process when targeting performance on the GPT-3.5 model.

> **Task: Causal Judgment**
> Provide causal attributions in complex scenarios by guiding the model to thoroughly analyze the critical steps, individual intentions, and specific actions that lead to outcomes. Emphasize the importance of identifying and prioritizing the primary cause in each scenario, focusing on direct causes rather than incidental factors. Define clear criteria for evaluating factors and determining the primary cause, considering the combined impact of multiple factors working in conjunction. Instruct the model to weigh the influence of various factors and explicitly guide it on handling conflicting actions and scenarios involving multiple individuals. Ensure that the model carefully considers all significant actions, intentions, and sequences of events leading to the final outcome to accurately attribute causation. Provide explicit instructions for distinguishing between direct causes and incidental factors, prioritizing immediate actions that directly influence outcomes. Define specific criteria for evaluating factors and determining the primary cause, especially in scenarios involving multiple individuals. Emphasize the need to analyze critical steps and actions leading to outcomes in order to accurately attribute causation.

> **Task: Geometric Shapes**
> Name the geometric shape accurately based on the provided SVG path. Carefully analyze the properties of the path, including the number of sides, angles, lengths of sides, and overall configuration, to determine the most appropriate geometric shape. Your options should encompass a wide variety of shapes, ranging from simple polygons to circles. Ensure that the model considers all relevant attributes before selecting the most suitable shape from the available options.
> Options: (A) circle, (B) equilateral triangle, (C) regular hexagon, (D) rhombus, (E) line segment, (F) octagon, (G) pentagon, (H) rectangle, (I) sector, (J) square, (K) trapezoid, (L) oval

**Task: Penguins**
Answer questions regarding the following tables of penguins and giraffes, ensuring to accurately reflect any changes made to the penguin table throughout our discussion. Please note these modifications specifically when determining key attributes such as age, weight, or when making comparisons between penguins and giraffes.

**Penguin Table:**
name, age, height (cm), weight (kg)
Louis, 7, 50, 11
Vincent, 9, 60, 11
Gwen, 8, 70, 15
(Any additions or deletions of penguins will be noted in subsequent questions)

**Giraffe Table:**
name, age, height (cm), weight (kg)
Jody, 5, 430, 620
Gladys, 10, 420, 590
Marian, 2, 310, 410
Donna, 9, 440, 650

For each question, provide clear and logical reasoning behind your answer. Remember to validate the latest state of the penguin table before responding, especially when involving comparisons with giraffes or assessing the attributes of the penguins.

Additionally, if modifications were made to the penguin table, please annotate them clearly in your response. This ensures that we maintain an accurate understanding of the current data.

---

**Task: Biosses**
Decide if these two sentences are (A) Not similar (B) Somewhat similar (C) Similar. Compare the specific regulatory mechanisms and molecular pathways mentioned in each sentence to determine their similarity, explicitly identifying the role of miRNA expression and binding, as well as the relevance of the molecular characteristics of GEFs and nucleotide-binding pockets in the context of the sentences. Analyze both the similarities and differences between the sentences, focusing on the nuances of the regulatory mechanisms and molecular pathways mentioned, and considering the implications for cancer types and cellular processes

---

**Task: CB**
What is the relationship between the following premise and the hypothesis? Premise: As the storm raged outside, with thunder clapping and lightning illuminating the dark sky, Sarah felt a wave of panic wash over her. She could hear the wind howling, and every crash of thunder made her heart race faster. Despite being tucked away under her thick blankets, she couldn't shake the feeling of terror that gripped her. The flickering candle nearby offered little comfort as she lay wide awake, listening to the chaos around her.
Hypothesis: Sarah felt a strong fear of the storm.
Entailment: The hypothesis is entailed by the premise. Sarah's panic and terror at the storm directly imply that she felt a strong fear of it. What is the relationship between the following premise and the hypothesis?
Options:
- Contradiction
- Neutral
- Entailment

Table 5: Average number of API calls required for prompt optimization and migration (lower is better), averaged over five runs. Percent changes are shown relative to the PromptAgent baseline: green indicates a reduction in calls, red indicates an increase. +FD = Feedback Diversification; +PR = Positive Reinforcement; BReAD = both combined. Despite leveraging more feedback signals, BReAD achieves faster convergence, reducing API calls by 0.5–3.3% in the standard setting (GPT-3.5) and by 4.2–6.2% in the migration setting (GPT-4o), demonstrating efficiency gains in optimization and transfer.

| Model | Method | Causal Judgment | Geometric Shapes | Penguins | Biosses | CB |
|---|---|---|---|---|---|---|
| GPT-3.5 | Baseline | 7670.4 | 11263.8 | 2663.6 | 3566.1 | 5489.5 |
| | Baseline+FD | 7918.8 (↑ 3.2%) | 11730.9 (↑ 4.1%) | 2735.4 (↑ 2.7%) | 3709.7 (↑ 4.0%) | 5606.2 (↑ 2.1%) |
| | Baseline+PR | 7039.8 (↓ 8.2%) | 10899.7 (↓ 3.2%) | 2533.3 (↓ 4.9%) | 3294.0 (↓ 7.6%) | 5337.2 (↓ 2.8%) |
| | BReAD | 7429.2 (↓ 3.1%) | 11204.0 (↓ 0.5%) | 2622.4 (↓ 1.5%) | 3449.1 (↓ 3.3%) | 5421.9 (↓ 1.2%) |
| GPT-4o | Baseline | 8575.8 | 9642.0 | 2980.4 | 3008.1 | 6228.9 |
| | Baseline+FD | 9271.0 (↑ 8.1%) | 10093.2 (↑ 4.7%) | 3068.3 (↑ 2.9%) | 3069.6 (↑ 2.0%) | 6393.6 (↑ 2.6%) |
| | Baseline+PR | 7963.2 (↓ 7.1%) | 9395.6 (↓ 2.6%) | 2908.9 (↓ 2.4%) | 2868.8 (↓ 4.6%) | 6079.1 (↓ 2.4%) |
| | BReAD | 8040.8 (↓ 6.2%) | 9049.4 (↓ 6.1%) | 2825.2 (↓ 5.2%) | 2860.1 (↓ 4.9%) | 5964.5 (↓ 4.2%) |

## D  Experimentation Costs

Table 5 reports the average number of API calls required by each method during prompt optimization and migration, for both GPT-3.5 and GPT-4o. Each value reflects the mean over five runs per task.

These figures serve as a proxy for real-world deployment costs, especially when interacting with commercial LLM APIs. While Feedback Diversification (+FD) increases the number of calls due to repeated querying, both Positive Reinforcement (+PR) and our full method, BReAD, achieve more efficient convergence. Notably, BReAD consistently reduces total API calls despite using more feedback samples per iteration, underscoring its sample efficiency and effective guidance.

Efficiency gains are more pronounced in the migration setting, where BReAD lowers API usage by 4.2–6.2% compared to the PromptAgent baseline. This cost reduction, combined with improved accuracy and stability, positions BReAD as a practical optimization strategy for scalable LLM-based systems.

