# OpenReview forum: "Rethinking Prompt Optimization: Reinforcement, Diversification, and Migration in Blackbox LLMs"
_TMLR — Rejected by TMLR_

### Review · Reviewer_cQT9 · 2025-08-20

**Summary Of Contributions:**

This paper introduces an automatic prompt optimization (APO) method for black-box LLMs.  Current methods primarily focus on error correction and neglect the valuable information. This paper proposes a method named Balanced Reinforcement and
Aggregated Diversification (BReAD) that aims to improve the performance of black-box LLMs by leveraging both positive and negative feedback signals. This paper also formally proposes Continual Prompt Optimization (CPO) to address the migration of prompts across different models. Empirical results demonstrate the effectiveness of the proposed methods. The strengths and weaknesses of the paper are as follows:

**Strengths:**
- This paper proposes the APO method in black-box LLMs, addressing both error correction and the utilization of valuable feedback signals.
- The introduction of Balanced Reinforcement and Aggregated Diversification (BReAD) provides a comprehensive framework for leveraging positive and negative feedback, enhancing the overall performance of prompt optimization.
- The formalization of CPO provides the migration of prompts across different model versions and providers.
- Empirical results demonstrate the effectiveness of the proposed methods.

**Weaknesses:**
- The clarity of the proposed methods and their distinctions from existing approaches could be improved, particularly in the methodology. Currently, it is difficult for readers to understand the whole process and the method used in the procedure.
- The paper mentions the formalization of CPO, which is missing in the methodology section. This is an overclaim.
- Baselines are not enough to support the claims made in the paper. The comparisons focus on PromptAgent (+FD/+PR variants); widely-used black-box APO baselines (e.g., APE/OPRO/ProTeGi) are not reported.
- LLMs are limited to GPT-3.5-turbo and GPT-4o, which is not enough to demonstrate the generality of the proposed methods.

**Audience:**

Yes

**Audience Explanation:**

- This paper proposes to improve the performance of black-box LLMs through automatic prompt optimization (APO) methods by utilizing both positive and negative feedback signals.
- This paper also introduces an interesting problem setting for migrating the prompts optimization from one large language model to another, which could have implications for future research in this area.

**Broader Impact Concerns:**

No broader impact concerns.

**Claims And Evidence:**

No

**Claims Explanation:**

- CPO formalization is missing. This paper states that Continual Prompt Optimization is “formally proposed,” but the methodology lacks formalization. Empirical study demonstrates the prompt migration from GPT-3.5-turbo to GPT-4o, but lacks the detailed explanation of the CPO.
- The methodology section is poorly organized and lacks details. It lacks information on how the modules are built and interact with each other.
- Baselines are not sufficient to support the claims made in the paper. The comparisons focus on PromptAgent (+FD/+PR variants); widely-used black-box APO baselines (e.g., APE/OPRO/ProTeGi) are not reported.
- Black-box LLMs are limited to GPT-3.5-turbo and GPT-4o, which is not enough to demonstrate the generality of the proposed methods.

**Requested Changes:**

- Please provide a detailed formalization of the Continual Prompt Optimization (CPO) in the methodology section.
- Can you elaborate more on the methodology section? You can start from the mathematical formulation of the problem and the modules.
- Can you provide more APO baselines to show the effectiveness of the proposed methods?
- Can you provide more black-box LLMs to demonstrate the generality of the proposed methods?

---

### Review · Reviewer_NGdE · 2025-08-24

**Summary Of Contributions:**

This paper builds a systematic framework in optimizing the prompt automatically. The framework contains 1) diversify the prompt candidates 2) evaluate the candidates and 3) optimize the prompts based on feedback.

Strengths (+) and Weaknesses (-) are:

+ The authors provide a systematic framework in formulating the prompt optimization. Which includes a brunch of potential popular prompt optimization methods
+ Experiments shows the improvement in GPT4o and 3.5.

- Although a series of methodologies are developed for the collecting the feedback, the prompt optimization itself is less discussed. It seems that the authors assumes that the APO methods can be used as regular optimization (e.g., SGD) algorithms, but actually these APO methods heavily depends on the prompt design and LLM version
- It's a little bit contradictory for the feedback generation. E.g., if an LLM call is strong enough to determine if a prompt is clear enough or not, the LLM call should be able to directly solve the question itself. Also, the it should be clarified that if the prompt is universally applied to all dataset or it's based on each individual questions.
- The current empirical is not complete. The authors should at least provide 1) the performance of the proposed algorithm in smaller models (7B, 13B) and 2) the migration between GPT4o to GPT3.5 and the migration from small model to large model. The current version only contain 3.5 result and migrate to 4o is not convincing.
- Regarding the different APO implementation, it seems that the authors does not specify which APO algorithm they choose. It's strongly recommended to include the performance for different APO algorithms for a better understanding of the framework.

**Audience:**

Yes

**Audience Explanation:**

LLM prompting is a growing topics, even there are some ongoing debates on if directly optimizing the prompts are helpful or not.

**Claims And Evidence:**

No

**Claims Explanation:**

- The current empirical is not complete. The authors should at least provide 1) the performance of the proposed algorithm in smaller models (7B, 13B) and 2) the migration between GPT4o to GPT3.5 and the migration from small model to large model. The current version only contain 3.5 result and migrate to 4o is not convincing.
- Regarding the different APO implementation, it seems that the authors does not specify which APO algorithm they choose. It's strongly recommended to include the performance for different APO algorithms for a better understanding of the framework.

Summary: experiments are not comprehensively support the claimed evidence.

**Requested Changes:**

- Could the author extend the experiments to 1) smaller models 2) more different migrations 3) include different APO methods
- Better justify the feedback generation, could the author justify that using another model to justify the promotion correctness is a reasonable idea? Could the author please clarify their assumption on individual prompts or global prompts?

---

### Review · Reviewer_id4x · 2025-08-30

**Summary Of Contributions:**

The paper focuses on the challenge of automatic prompt optimization (APO) for black-box large language models (LLMs), highlighting limitations of current APO methods that center feedback on correcting errors while ignoring positive aspects of correct predictions. The authors introduce a new APO framework, BReAD, that frames feedback as both negative and positive reinforcement and adds feedback diversification to aggregate multiple feedback signals, reducing noise. Importantly, the paper formalizes Continual Prompt Optimization (CPO), allowing optimized prompts to be efficiently migrated between LLMs (e.g., from GPT-3.5-turbo to GPT-4o), countering degradation seen in naive migration. Empirical evaluation across diverse tasks shows accuracy, convergence, and efficiency gains over strong baselines, particularly in migration settings.

**Audience:**

Yes

**Audience Explanation:**

The main claims in this paper are: (1) enhancing prompt optimization by using both positive and negative reinforcement leads to more effective retention of useful instructions, (2) feedback diversification increases robustness by filtering noisy or contradictory feedback, and (3) the framework enables more successful prompt migration across different LLMs, mitigating the detrimental effects of naive migration. These claims are supported by experimental evidence, which demonstrates statistically significant improvements in mean accuracy.

**Broader Impact Concerns:**

No ethical concerns specific to this work.

**Claims And Evidence:**

No

**Claims Explanation:**

There are no formal theoretical theorems or proofs in the paper. Theoretical motivation is provided primarily through the analogy of negative/positive reinforcement and by relating prompt optimization to reinforcement learning and continual learning paradigms, but all justifications are informal.

**Requested Changes:**

1. Lack of comparisons with key, directly related prior APO frameworks that use RL or sequential decision processes (e.g., RLPrompt, StablePrompt, GRL-Prompt, AutoPDL).

2. Limited exploration of prompt migration between truly different architectures/providers or with more adversarial transfer scenarios.

3. Few qualitative analyses—for instance, showing example prompts before/after optimization, or concrete LLM behaviors that fail/succeed under different feedback regimes.

4. Hyperparameter sensitivity, while discussed, could use more thorough automated or cross-task tuning.
Real-world deployment experiments are missing, but the paper does address efficiency via API usage.

Overall, the analysis is genuinely informative but could be strengthened through additional baselines, examples, and expanded error inspection.

---

### Decision · Action_Editor_TK5t · 2025-10-13

**Recommendation:** Reject

**Additional Comments:**

NA

**Audience:**

Yes

**Audience Explanation:**

LLM prompting is an emerging topic. The proposed method and experiments in this paper, if supported by further evidence, would draw attention from the audience of this journal.

**Claims And Evidence:**

No

**Claims Explanation:**

This paper introduces an automatic prompt optimization (APO) method for black-box LLMs. Existing methods primarily focus on error correction while neglecting the valuable information contained in feedback signals. To address this, the paper proposes Balanced Reinforcement and Aggregated Diversification (BReAD), a method designed to improve black-box LLM performance by leveraging both positive and negative feedback. In addition, the paper formally introduces Continual Prompt Optimization (CPO) to handle prompt migration across different models.

However, the analysis presented in the paper is mostly informal. Despite the claims of theoretical grounding, no clear formalization of the method is provided. The only supporting evidence comes from experiments, which are not sufficiently comprehensive. As noted by the reviewers, several improvements are needed to strengthen the work: (1) exploring additional mitigation strategies, such as varying model architectures, model sizes, or adversarial settings; (2) including comparisons with key, directly related APO frameworks that employ reinforcement learning or sequential decision processes (e.g., RLPrompt, StablePrompt, GRL-Prompt, AutoPDL); and (3) conducting hyperparameter sensitivity analyses.

Unfortunately, the authors did not provide a response or revision, even after being granted an extension. Therefore, the paper is rejected in its current form.